# Energy-Free Guidance of Geometric Diffusion Models for 3D Molecule Inverse Design

**Sanjay Nagaraj** [1] [*]  **Jiaqi Han** [1] [*]  **Aksh Garg** [1] [*]  **Minkai Xu** [1]

## Abstract

Molecule inverse design is of critical significance in drug discovery which requires molecules to be generated based on certain chemical properties or structural compositions. Generative models, most popularly diffusion models, have shown great promise in performing inverse design through conditioning techniques and/or explicit energy guidance during sampling. In this work, we propose a novel guidance framework, Energy-Free Guidance for Geometric Diffusion Models (EFG-GDM), that effectively boosts the utility of molecule inverse design without any auxiliary energy head for guidance. The key innovation lies in the joint training strategy for the conditional and unconditional score models via random masking, which are then composed during sampling in an SE(3)-equivariant fashion, ensuring the critical physical symmetry of the geometric distribution. This feature alleviates practitioners from needing additional efforts in training energy prediction heads and avoids the adversarial gradient coming from them. We conduct experiments on a diverse range of inverse design tasks on QM9, showing that our approach achieves state-of-the-art on 4 out of 6 design targets without leveraging any external energy gradients. Code is available at https://github.com/akshgarg7/cfgedm.

## 1. Introduction

Generative models, especially diffusion models (Sohl-Dickstein et al., 2015; Song & Ermon, 2019; Ho et al., 2020; Song et al., 2021), have made remarkable progress

---

[*]Equal contribution . Jiaqi and Minkai discussed Energy-free Guidance and Jiaqi explored the idea; Independently, Aksh and Sanjay studied classifier-free guidance as a Stanford CS236 course project mentored by Minkai. All authors later collaborated together to wrap up individual results as this paper. [1]Department of Computer Science, Stanford University, Stanford, CA 94305, USA. Correspondence to: Minkai Xu <minkai@stanford.edu>.

*Accepted at the 1st Machine Learning for Life and Material Sciences Workshop at ICML 2024.* Copyright 2024 by the author(s).

in *de novo* 3D molecular design, a fundamental component in the blueprint of artificial intelligence-driven drug discovery (Cheng et al., 2021). The core to the success of these models, namely geometric diffusion models (Xu et al., 2022; 2023; Hoogeboom et al., 2022), lies in the preservation of the physical symmetry in the marginal distribution of the generated molecules. In particular, they feature an SE(3)-equivariant diffusion process with the reverse kernels parameterized by SE(3)-equivariant graph neural networks, thus ensuring that the molecular structures reachable via rotations and translations in 3D share the same likelihood. This consideration has been demonstrated as being effective in promoting the sample quality of geometric diffusion models.

Among the tasks in molecule design, 3D molecular *inverse design* (Bao et al., 2022) has attracted growing interest, which requires generation of 3D molecules conditioning on desired properties, *e.g.*, polarizability or structural patterns. However, tackling such a task is highly challenging, since the inverse design requires the model to capture gradient signals that guide the generated molecule towards the desired property. Moreover, the target distribution for the generated molecule should also preserve the SE(3)-invariance. To address these challenges, existing works resort to explicitly training a conditional geometric diffusion model that takes as input the property to design (Hoogeboom et al., 2022; Xu et al., 2023) or employing an externally trained energy prediction model as the source of the gradient for guidance (Bao et al., 2022). Yet, conditional models have been reported to have relatively weak guidance towards the target distribution, leading to unsatisfactory designed samples that fail to closely abide by the target property. The energy guidance technique (Bao et al., 2022), although offering stronger guidance, requires additional efforts in training an energy prediction model per property, and is applied as an adversarial gradient during sampling, which loses guarantees on sampling quality.

To overcome these challenges, in this work, we propose energy-free guidance for geometric diffusion models (EFG-GDM) for 3D molecule inverse design, inspired by classifier-free diffusion guidance (Ho & Salimans, 2022). Our key innovation lies in training an SE(3)-equivariant score model that can produce both the unconditional and the conditional

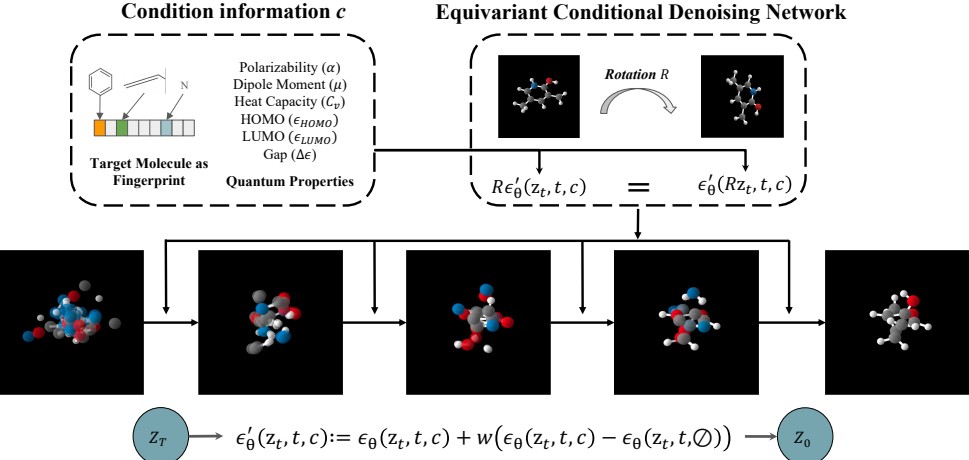

Figure 1: Schematic of EFG-GDM. EFG-GDM generates molecules under a given condition $c$ and has an equivariant score network that uses conditional and unconditional scores for guidance.

score by randomly masking the conditioning property during training. The model is then leveraged in sampling, which enforces guidance through the composition of the unconditional and conditional score, controlled by a guidance weight. By such design, our approach gets rid of training additional energy prediction heads and derives the guidance through the scores without any application of an adversarial gradient. The entire framework is also carefully designed to meet the physical symmetry, *e.g.*, the SE(3)-invariance of the marginal distribution, when the guidance is applied.

We comprehensively benchmark EFG-GDM on multiple inverse design tasks on the QM9 dataset (Ramakrishnan et al., 2014). Our results illustrate that EFG-GDM achieves state-of-the-art performance in terms of the mean-absolute-error between the property of the generated molecule and the design target on 4 out of 6 properties, while matching the performance of the energy guidance approach on the others with less degradation to molecule stability. We also perform ablation studies that further verify the necessity of the joint training strategy via masking, as opposed to leveraging both a conditional and unconditional score model.

## 2. Related Work

**Generative modeling for molecule design.** There have been initial attempts of generative modeling for molecule design using variational autoencoders (Lim et al., 2018), generative adversarial networks (Lee et al., 2021), and flow-based models (Luo & Ji, 2022; Satorras et al., 2021a). Recently, geometric diffusion models have emerged and achieved superior performance in 3D molecule generation. In particular, Hoogeboom et al. (2022) propose EDM, an equivariant diffusion model with symmetry constraints imposed on the transition kernels and marginal distribution. Xu et al. (2023) further incorporate a latent diffusion mechanism for enhanced performance. Our approach can be seamlessly

built upon these geometric diffusion models, but with additional enhancements that offer stronger guidance to match the inverse design target.

**Diffusion guidance.** Diffusion guidance aims to drive the marginal distribution towards a certain property. Classifier guidance (Song et al., 2020; Dhariwal & Nichol, 2021; Zhao et al., 2022) was initially proposed to improve sample quality for conditional image generation. Energy guidance has been explored for molecule inverse design through an invariant energy function to ensure equivariance (Bao et al., 2022). However, this approach requires training additional energy predictors and is applied as an adversarial gradient to the sampling process, degrading the sample quality. Classifier-free guidance (CFG) (Ho & Salimans, 2022), instead, derives the guidance through a composition of the conditional and unconditional score, both of which are computed from one score network. It has been widely adopted on tasks like image generation due to its simplicity and desirable performance. Our approach is inspired by CFG but is developed for 3D molecule inverse design with additional considerations in ensuring the SE(3)-invariance of the marginal.

## 3. Method

### 3.1. Preliminaries

**Notations.** A molecule is represented as $(\mathbf{h}, \mathbf{x})$ where $\mathbf{h} \in \mathbb{R}^{N \times H}$ is the one-hot encoding of the atomic numbers, and $\mathbf{x} \in \mathbb{R}^{N \times 3}$ is the coordinate with $N$ as the number of atoms.

**Geometric diffusion models.** We mainly focus on Equivariant Diffusion Models (EDM) (Hoogeboom et al., 2022) in this work, while our framework can also be seamlessly incorporated into other geometric diffusion models like geometric latent diffusion (Xu et al., 2023). Specifically, EDM constructs a diffusion process on the product space spanned by invariant and equivariant features with the la-

tent variables $\mathbf{z}_t := [\mathbf{z}_t^{\mathrm{x}}, \mathbf{z}_t^{\mathrm{h}}]$ for $t = 1, \cdots, T$, where $T$ is the number of diffusion steps, $\mathbf{z}_t^{\mathrm{x}} \in \mathbb{R}^{N \times 3}$ is the latent variable for equivariant feature, and $\mathbf{z}_t^{\mathrm{h}} \in \mathbb{R}^{N \times H}$ is similarly defined for the invariant feature. The framework consists of a forward diffusion and a reverse denoising process, both of which are Markovian. The forward diffusion process is given by the transition $q(\mathbf{z}_t|\mathbf{x}, \mathbf{h}) = \mathcal{N}(\mathbf{z}_t^{\mathrm{x}}; \alpha_t\mathbf{x}, \sigma_t^2\mathbf{I}) \cdot \mathcal{N}(\mathbf{z}_t^{\mathrm{h}}; \alpha_t\mathbf{h}, \sigma_t^2\mathbf{I})$, where $\sigma_t$ is specified by certain noise schedule (Ho et al., 2020). In correspondence, the reverse denoising process is specified by the transition kernel $p_\theta(\mathbf{z}_{t-1}|\mathbf{z}_t) := \mathcal{N}(\mathbf{z}_{t-1}; \boldsymbol{\mu}_\theta(\mathbf{z}_t, t), \rho_t^2\mathbf{I})$, where $\boldsymbol{\mu}_\theta(\mathbf{z}_t, t)$ is parameterized by an EGNN (Satorras et al., 2021b), and $\rho_t$ is also predefined. With the equivariant reverse kernel and the isotropic Gaussian prior, the marginal is guaranteed rotation-invariance. To deal with translation-invariance, all variables $\mathbf{z}_t$ are projected on the translation-invariant zero-CoM (center-of-mass) subspace (more details in Appendix A). The model is optimized by the variational lower bound, which is equivalent to the simplified noise-prediction objective in (Hoogeboom et al., 2022):

$$\mathcal{L} = \mathbb{E}_{(\mathbf{x},\mathbf{h})\sim\mathcal{D}_{\mathrm{data}}, t\sim\mathrm{Unif}(1,T), \boldsymbol{\epsilon}\sim\mathcal{N}(\mathbf{0},\mathbf{I})} \left[\lambda(t)\|\boldsymbol{\epsilon} - \boldsymbol{\epsilon}_\theta(\mathbf{z}_t, t)\|^2\right], \quad (1)$$

where $\lambda(t)$ is the weighting factor and $\boldsymbol{\epsilon}_\theta$ is the score network (Song et al., 2021), which is a re-parameterization of the mean $\boldsymbol{\mu}_\theta$ in the reverse kernel.

## 3.2. Energy-free Guidance for Geometric Diffusion Models

**Conditioning.** In order to perform inverse design, Hoogeboom et al. (2022); Xu et al. (2023) train a conditional geometric diffusion model that utilizes the design target as an explicit condition. This is fulfilled by concatenating the design target $c$ to the input of the score network, leading to the following conditional training objective:

$$\mathcal{L}_{\mathrm{cond}} := \mathbb{E}_{(\mathbf{x},\mathbf{h},c),t,\boldsymbol{\epsilon}} \left[\lambda(t)\|\boldsymbol{\epsilon} - \boldsymbol{\epsilon}_\theta(\mathbf{z}_t, c, t)\|^2\right], \quad (2)$$

where $(\mathbf{x}, \mathbf{h}, c) \sim \mathcal{D}_{\mathrm{data}}, t \sim \mathrm{Unif}(1, T)$ and $\boldsymbol{\epsilon} \sim \mathcal{N}(\mathbf{0}, \mathbf{I})$.

**Energy-guidance.** Bao et al. (2022) reveal that the simple conditioning technique is insufficient to offer strong guidance towards the design target. To promote the design efficacy, they additionally train an energy prediction network $\varphi_\eta$ for each design target $c$ via the loss,

$$\mathcal{L}_{\mathrm{EG}} = \mathbb{E}_{(\mathbf{x},\mathbf{h},c)\sim\mathcal{D}_{\mathrm{data}}, t\sim\mathrm{Unif}(1,T)}\|\varphi_\eta(\mathbf{z}_t, t) - c\|^2. \quad (3)$$

The energy network $\varphi_\eta$ is then leveraged in sampling to provide additional guidance over the original score:

$$\boldsymbol{\epsilon}_\theta'(\mathbf{z}_t, c, t) := \boldsymbol{\epsilon}_\theta(\mathbf{z}_t, c, t) \\ - s[\mathrm{Proj}\left(\nabla_{\mathbf{z}_t^{\mathrm{x}}}\varphi_\eta(\mathbf{z}_t, t)\right), \nabla_{\mathbf{z}_t^{\mathrm{h}}}\varphi_\eta(\mathbf{z}_t, t)], \quad (4)$$

where $\mathrm{Proj}(\cdot)$ is the projection on the zero-CoM subspace for translation invariance and $s$ is the guidance weight. Note that the guidance is applied on the conditional model, which is reported to yield better performance.

**Energy-free guidance.** Here we adopt a different strategy, *i.e.*, energy-free guidance, that does not require additionally training the energy predictor, and does not inject the external energy gradient in Eq. 4 as an adversarial gradient term. Inspired by classifier-free diffusion guidance (Ho & Salimans, 2022), we provide the guidance as the inherent difference in the conditional and unconditional score. To obtain these scores without introducing additional parameters or external networks, we model the conditional and unconditional scores in one network, which is optimized by the following objective:

$$\mathcal{L}_{\mathrm{EFG}} := \mathbb{E}_{m\sim\mathrm{Bern}(1;\mathrm{p}),(\mathbf{x},\mathbf{h},c),t,\boldsymbol{\epsilon}}\lambda(t) \\ \left[m\|\boldsymbol{\epsilon} - \boldsymbol{\epsilon}_\theta(\mathbf{z}_t, t, c)\|^2 + (1-m)\|\boldsymbol{\epsilon} - \boldsymbol{\epsilon}_\theta(\mathbf{z}_t, t, \varnothing)\|^2\right], \quad (5)$$

where similarly $(\mathbf{x}, \mathbf{h}, c) \sim \mathcal{D}_{\mathrm{data}}, t \sim \mathrm{Unif}(1, T), \boldsymbol{\epsilon} \sim \mathcal{N}(\mathbf{0}, \mathbf{I})$, $m \sim \mathrm{Bern}(1; \mathrm{p})$ is the Bernoulli distribution that produces the sample 1 with probability $p$ and 0 with probability $1 - p$, and the symbol $\varnothing$ represents the indicator for the unconditional score, which is simply implemented as an all-zero vector. In practice, we first sample the mask $m$ from the Bernoulli distribution and then decide which term is activated in Eq. 5 to avoid computing two scores in each forward pass. With the score network properly trained, during sampling we compose the conditional and unconditional scores obtained from the score network to exert the guidance:

$$\boldsymbol{\epsilon}_\theta'(\mathbf{z}_t, t, c) := \boldsymbol{\epsilon}_\theta(\mathbf{z}_t, t, c) + w\left(\boldsymbol{\epsilon}_\theta(\mathbf{z}_t, t, c) - \boldsymbol{\epsilon}_\theta(\mathbf{z}_t, t, \varnothing)\right), \quad (6)$$

where $w$ is the guidance weight. The adjusted score $\boldsymbol{\epsilon}_\theta'(\mathbf{z}_t, t, c)$ is then utilized in the original sampling procedure as a replacement of the original score $\boldsymbol{\epsilon}_\theta(\mathbf{z}_t, t, c)$. Remarkably, with the proposed design, we are still able to guarantee the rotation-equivariance and translation-invariance of the score after applying guidance, *i.e.*, $\boldsymbol{\epsilon}_\theta'(\mathbf{R}\mathbf{z}_t, t, c) = \mathbf{R}\boldsymbol{\epsilon}_\theta'(\mathbf{z}_t, t, c)$, which is vital to guarantee the SE(3)-invariance of the marginal distribution, hence that of the sampled molecules.

## 4. Experiments

**Task and metrics.** We employ the QM9 dataset which consists of quantum properties and structural coordinates for approximately 130,000 molecules, each with up to nine heavy atoms from the group C, N, O, F. We evaluate EFG-GDM on the single-property inverse design task, where given an input property value, our model constructs a molecular structure with that property. We test on 6 standard QM properties: polarizability ($\alpha$), dipole moment ($\mu$), heat capacity

Table 1: Performance comparison on conditional generation tasks. Results of baselines are taken from Bao et al. (2022).

| | $C_v$ (cal/mol) | | | | $\alpha$ (Bohr$^3$) | | |
|---|---|---|---|---|---|---|---|
| | MAE | Atom Stable | Mol Stable | | MAE | Atom Stable | Mol Stable |
| CondEDM | 1.065 | 98.25 | 80.82 | CondEDM | 2.78 | 98.13 | 79.33 |
| EEGSDE ($s$=10) | 0.941 | 98.03 | 79.07 | EEGSDE ($s$=3) | 2.50 | 98.26 | 80.95 |
| EFG-GDM ($w$=1.5) | **0.902** | 97.93 | 77.01 | EFG-GDM ($w$=3) | **2.27** | 98.58 | 90.10 |

| | $\Delta_\epsilon$ (meV) | | | | $\epsilon_{\text{LUMO}}$ (meV) | | |
|---|---|---|---|---|---|---|---|
| | MAE | Atom Stable | Mol Stable | | MAE | Atom Stable | Mol Stable |
| CondEDM | 671 | 98.30 | 81.95 | CondEDM | 601 | 98.26 | 81.34 |
| EEGSDE ($s$=3) | 487 | 97.76 | 77.43 | EEGSDE ($s$=3) | 447 | 98.14 | 80.00 |
| EFG-GDM ($w$=3.5) | **423** | 97.68 | 83.75 | EFG-GDM ($w$=2.5) | **410** | 98.54 | 90.63 |

| | $\mu$(D) | | | | $\epsilon_{\text{HOMO}}$(meV) | | |
|---|---|---|---|---|---|---|---|
| | MAE | Atom Stable | Mol Stable | | MAE | Atom Stable | Mol Stable |
| CondEDM | 1.123 | 98.17 | 80.25 | CondEDM | 371 | 98.17 | 79.61 |
| EEGSDE ($s$=2) | **0.777** | 98.06 | 78.92 | EEGSDE ($s$=1) | **302** | 98.08 | 78.90 |
| EFG-GDM ($w$=4.25) | 0.790 | 98.34 | 75.00 | EFG-GDM ($w$=1.0) | 318 | 98.59 | 81.25 |

Table 2: Ablation study on the joint training strategy.

| | $\Delta_\epsilon$ (meV) | | $\epsilon_{\text{LUMO}}$(meV) | | $\alpha$ (Bohr$^3$) | |
|---|---|---|---|---|---|---|
| | MAE | Mol Stable | MAE | Mol Stable | MAE | Mol Stable |
| EFG-GDM | **474** | 83.75 | **410** | 90.63 | **2.27** | 90.10 |
| EFG-GDM-sep | 499 | 77.50 | 464 | 77.44 | 2.64 | 61.60 |

($C_v$), highest orbital energy ($\epsilon_{\text{HOMO}}$), lowest orbital energy ($\epsilon_{\text{LUMO}}$) and their gap ($\Delta\epsilon$). We train a model and a property-prediction network for each property. Post-training, given a range of property values $c$ we sample molecules from our generative model and use the pretrained property predictor to estimate their property values $\hat{c}$. We calculate the MAE between $c$ and $\hat{c}$ over all generated molecules per property. We also report the mean molecular stability of our molecules to ensure stability/structure is not heavily compromised. More details are deferred to Appendix B.

**Baselines.** We compare against conditional EDM ((Hoogeboom et al., 2022)) and the energy-guided approach EEGSDE ((Bao et al., 2022)). EEGSDE reports results over a range of guidance weights. Similarly, we range our guidance weight between $w = 0$ to $w = 5$, and select the best reported results from our EFG-GDM and EEGSDE according to MAE. We also report our result swept over the range of varied guidance weights in Appendix C.2.

**Main results.** The main results are presented in Table 1. Remarkably, our approach outperforms CondEDM and EEGSDE, the current state-of-the-art method, on 4 out of 6 tasks: $\alpha$, $\epsilon_{\text{LUMO}}$, $C_v$, and $\Delta\epsilon$. More concretely, we decrease the MAE by 6.8%, 4.1%, 4.0%, and 2.7% for the $\alpha$, $\epsilon_{\text{LUMO}}$, $C_v$, and $\Delta\epsilon$ tasks relative to EEGSDE and 15.1%, 34%, 29.4%, and 15.3% relative to CondEDM. In particular, we observe that EEGSDE is able to improve MAE over CondEDM with the aid of the energy gradient. However, our EFG-GDM produces molecules that align with the design target more closely and all scores can be computed by one score network without the need for training an additional energy predictor. Interestingly, while achieving lower MAE, EFG-GDM generally preserves higher stability,

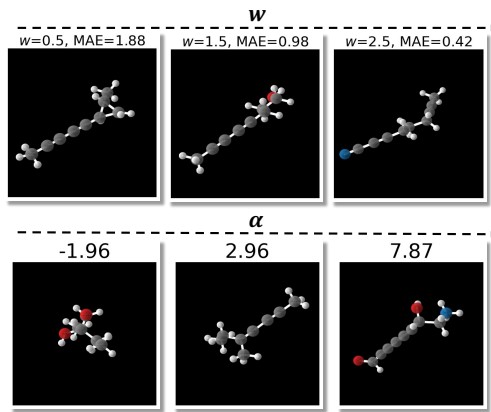

Figure 2: *Top*: Generated molecules on QM9 when conditioned on $\alpha$ with increasing guidance weight $w$. *Bottom*: Impact of increasing $\alpha$. Molecules get increasingly polar.

and even enhances the stability by a large margin on $\alpha$ and $\epsilon_{\text{LUMO}}$. By contrast, EEGSDE usually incurs degradation in stability, due to the adversarial gradient in the predictor.

**Ablations and analyses.** We further ablate our design in the training strategy with results depicted in Table 2. In detail, we implement a variant of EFG-GDM that separately trains two score networks, one for the conditional score, and the other for the unconditional score. We observe that our joint training strategy with random masking consistently improves the performance, both for the MAE and molecule stability. We speculate the joint training approach helps inherently align the conditional and unconditional score in the direction of generating stable molecules, which makes their difference better reflect the direction of the guidance towards the design target. By contrast, separately training two networks may result in more gradient conflicts due to the disentanglement in the parameter space.

We further illustrate how the guidance weight influences generation at the top of Fig. 2. We fix our original noise and run our model deterministically with guidance weights

ranging from 0.5 to 2.5. As $w$ increases, we see that the generated structure matches the target property more closely (as evidenced by the lower MAE), highlighting that our guidance empirically works as expected. We provide more examples of varying guidance weights in Appendix C.2. Additionally, we demonstrate that our model is responsive to input property values. Specifically, as we range the polarizability $\alpha$ from $-1.96$ to $7.87$ at the bottom of Figure 2, we notice that our generated molecules go from being more closed and ring-like to being more elongated and having noticeably more polar chains.

## 5. Conclusion

In this work, we introduce energy-free guidance for geometric diffusion models (EFG-GDM), with applications to 3D molecule inverse design. The core insight is to train a score network that can produce conditional and unconditional scores, and use their difference as the inherent guidance signal. Experiments demonstrate EFG-GDM achieves superior performance on QM9 inverse design tasks. Future work include extending EFG-GDM to more advanced geometric diffusion models and applying it to more datasets and tasks, *e.g.*, target-aware molecule generation.

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

## A. Translation-Invariant Zero-CoM subspace

The importance of translational and rotational invariance in 3D molecular modeling is acknowledged in several works (Köhler et al., 2019; Xu et al., 2022). However, it is impossible for a properly normalized distribution in the ambient space $\mathbb{R}^{N \times 3}$ to be translation-invariant (Satorras et al., 2021a; Hoogeboom et al., 2022). A widely adopted workaround is to restrict the distribution into a translation-invariant subspace, among which the zero center-of-mass (CoM) is the most popular choice due to the simplicity. In detail, the zero-CoM subspace is given by $\{\mathbf{x} : \sum_{i=1}^{N} \mathbf{x}_i = \mathbf{0}\}$. For any data point $\mathbf{x}$, the projection of it onto the zero-CoM subspace is given by $P(\mathbf{x}) = \mathbf{x} - \frac{1}{N} \sum_{i=1}^{N} \mathbf{x}_i$. The Gaussian restricted in the zero-CoM subspace is also isotropic and thus rotation-invariant. To sample from it, one can directly sample from the Gaussian in the ambient space, and then project the sample onto the subspace via the projection $P(\cdot)$. In practice, geometric diffusion models operate on the product space of the equivariant feature $\mathbf{x}$ and invariant feature $\mathbf{h}$, where the former is restricted to the zero-CoM subspace, implying that all the latent variables $\mathbf{z}_t^{\mathbf{x}}$ reside in the zero-CoM subspace, fulfilled by repeatedly applying the projection during sampling. We refer the readers to Satorras et al. (2021a); Bao et al. (2022) for more details.

## B. Additional Experimental Details

Following Hoogeboom et al. (2022); Bao et al. (2022), we partition QM9 into training, validation, and test sets with 100,000, 18,000, and 13,000 molecules, respectively. The training set is further split into two halves: the first half is used to train a property prediction network, which is used to evaluate the alignment of the generated molecules with their targeted properties. The second half is used to train the diffusion models.

We also follow the hyperparameter setup as described in Hoogeboom et al. (2022) for both the model architecture and model training.

## C. Additional Results

### C.1. Effect of guidance weight on MAE and molecular stability

Table 3 presents additional results from guidance sweeps. As we increase $w$, we observe that there is a tradeoff between our MAEs and molecular stability. This is evident for $\mu, \alpha,$ and $\epsilon_{LUMO}$, where at low $w$, the molecular stability is around 0.90, which is competitive with most models for unconditional generation. In fact, even with $w = 0.75$, our $\alpha$ model's MAE is still state-of-the-art relative to other conditional models considered in this work. This offers additional evidence to support that our masking approach does in fact improve molecular generation, as highlighted by the high molecular stabilities, and that our guidance allows us to trade stability for MAE, as desired.

Table 3: Additional results on the MAE and the molecule stability (MS) of generated molecules targeted to $\mu$, $\alpha$, and $\epsilon_{LUMO}$.

| $\mu$ | | | $\alpha$ | | | $\epsilon_{\text{LUMO}}$ | | |
|---|---|---|---|---|---|---|---|---|
| Config | MAE | Mol Stable | Config | MAE | Mol Stable | Config | MAE | Mol Stable |
| EFG-GDM (w=0) | 1.06 | **0.91** | EFG-GDM (w=0.75) | 2.45 | **0.90** | EFG-GDM (w=0.5) | 539 | **0.89** |
| EFG-GDM (w=2) | 0.93 | 0.86 | EFG-GDM (w=2.5) | 2.4 | 0.85 | EFG-GDM (w=2) | 437 | 0.87 |
| EFG-GDM (w=4.25) | **0.79** | 0.75 | EFG-GDM (w=3) | **2.27** | 0.90 | EFG-GDM (w=2.5) | **410** | 0.91 |

## C.2. Visualizations of Generated Molecules Under Increasing Guidance

Figure 3 demonstrates effects of increasing guidance on all 6 quantum properties. As expected, as $w$ increases, the MAE between the generated and target properties decreases.

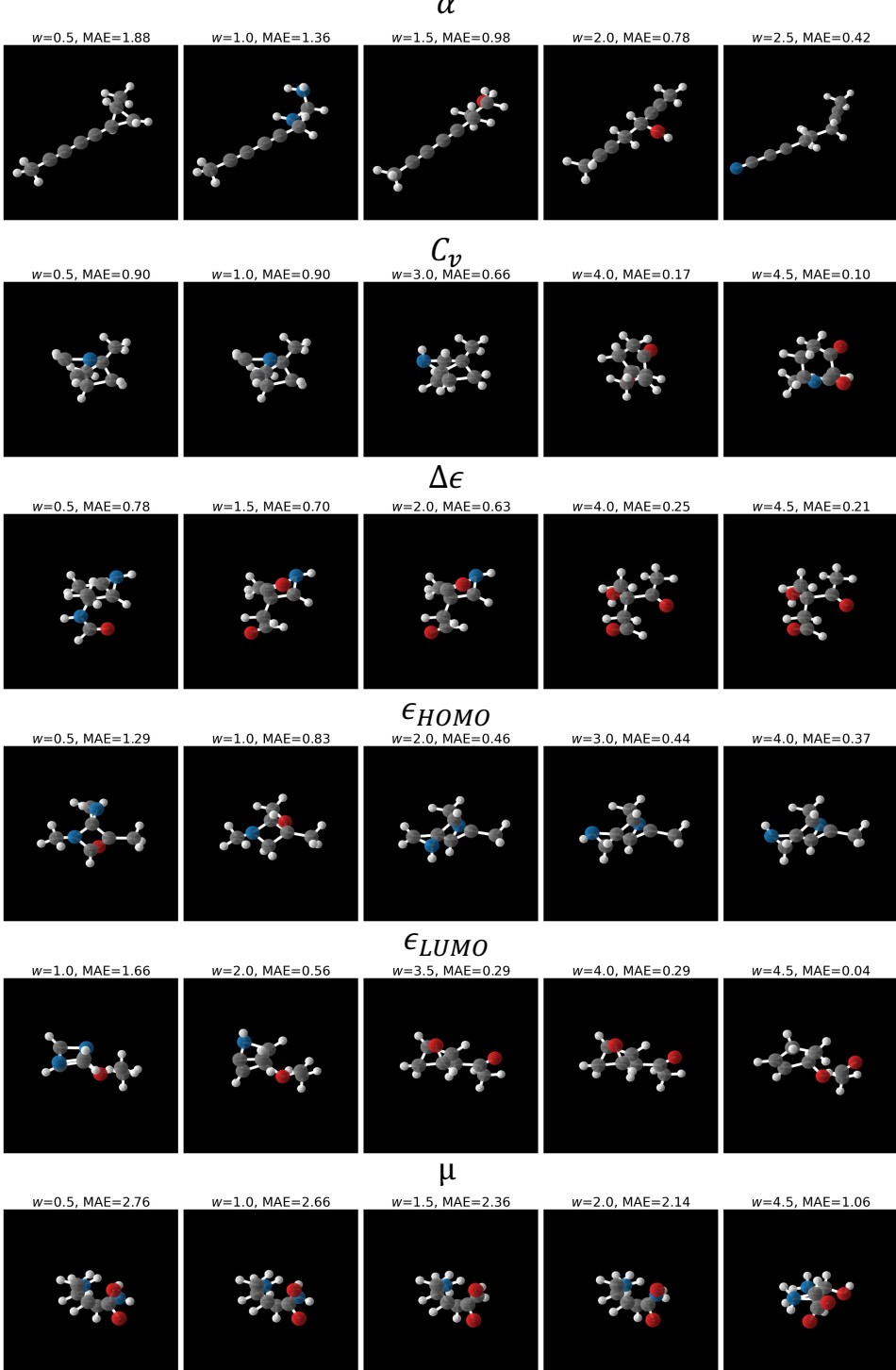

Figure 3: Impact of increasing varying our guidance weight, $w$ on all 6 properties.

## C.3. Visualizations of Generated Molecules Under Increasing Property Value

Figure 4 demonstrates effects of increasing the target property on all 6 quantum properties. As expected, as the property weight increases, the designed molecules exhibit that property more strongly.

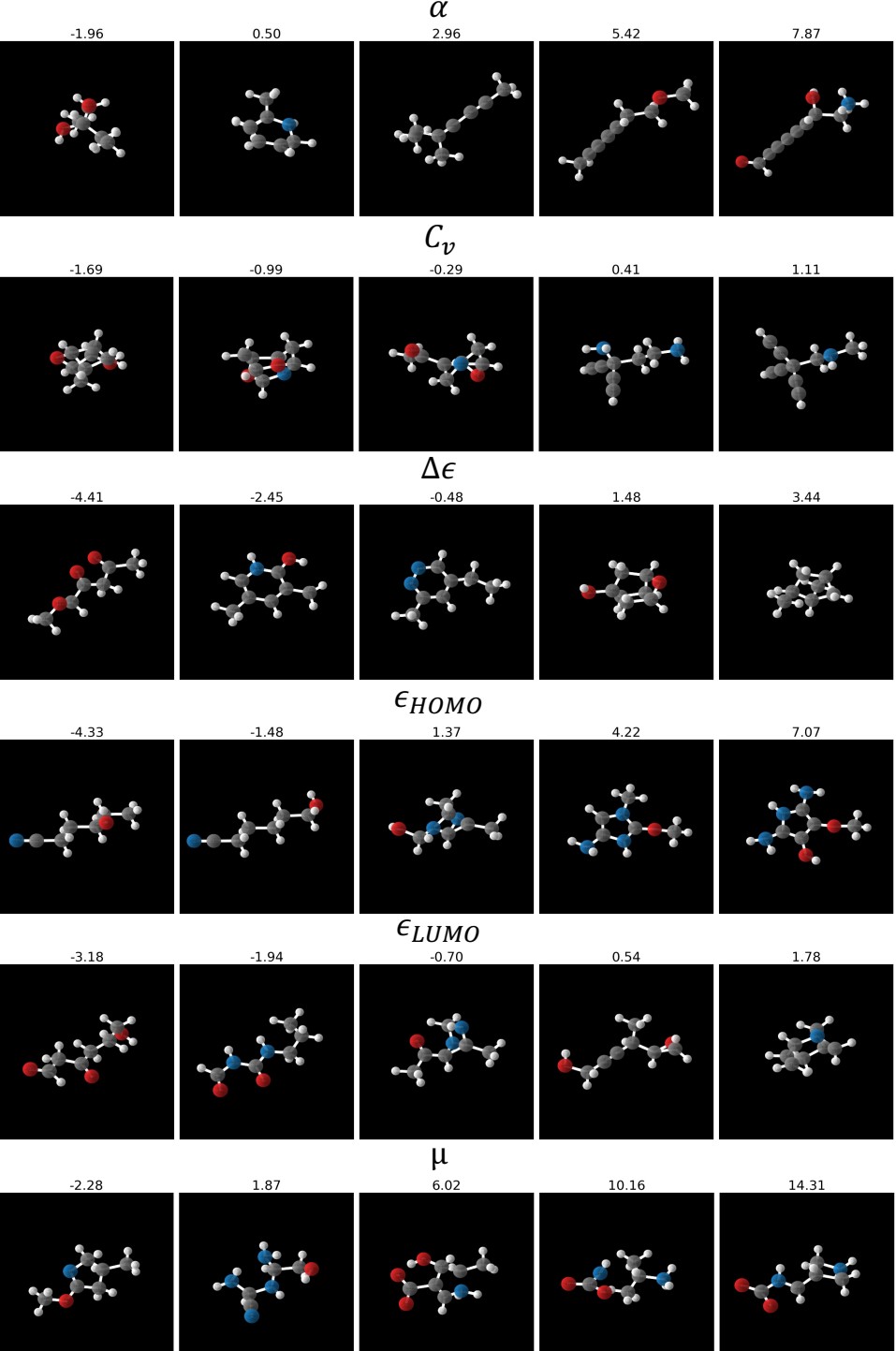

Figure 4: Impact of increasing property values on designed molecules across QM9 properties.