# OpenReview forum: "Energy-Free Guidance of Geometric Diffusion Models for 3D Molecule Inverse Design"
_ICML.cc/2024/Workshop/ML4LMS — ML4LMS Poster_

### Official Review · Reviewer_2VEw · 2024-06-05
**Interesting idea, benchmarking could be more extensive**

**Rating:** 6
**Confidence:** 2

**Review:**

Pros:
- Clearly presented work

Cons:
- Benchmarking against more models, also more recent models e.g GaUDI https://www.nature.com/articles/s43588-023-00532-0 would be beneficial

---

### Official Review · Reviewer_KyUC · 2024-06-11
**Clever improvements to score methods, but more extensive evaluation of results needed.**

**Rating:** 4
**Confidence:** 4

**Review:**

**Summary**.
"Energy Free Guidance of Geometric Diffusion Models for 3D Molecule Inverse Design"  describes a geometric diffusion model for 3D molecule inverse design which utilizes combined conditional and unconditional score models.

**Strengths**.
1. *Performance:* the property improvements provided by EFG-GDM is  stronger than those provided by conditional models (CondEDM) for all six properties tested.
2. *Efficiency:* previous approaches utilizing energy based oracles require the development and training of an energy prediction model for each conditioned property. Using the combined score models, EFG-GDM is able to produce lower MAE than EEGSDE (an energy based model) in four of six evaluation metrics.

**Weaknesses**.
1. *Innovation:* the paper describes a method which has modest differences from existing publications (Bao et al and Hoogeboom et al), who describe energy based and conditional approaches, respectively. The improvement of the model is also relatively limited over existing models (and in some cases not an improvement), and is evaluated on a well known molecular dataset (QM9), whose molecules are relatively limited in size.
2. *Utility:* the conditional generation tasks are only for physical and/or chemical properties. In many cases, it can be assumed that a user would want to control the molecular shape and positions of hydrophobic, and hydrogen bond donors/acceptors. Indeed, in many cases, the molecule shape appears to have changed considerably (e.g. many of the row-wise progressions in figure 4 in the appendix). Such an oracle would likely be much more difficult to create. Lack of control over molecular structure is a significant limitation, in this reviewer's opinion.
3. *Evaluation of Results:* the quality of the generated molecules was not assessed from a structural standpoint. And many of the molecules appear to have aliphatic chains whose bond angles are near 180 degrees, which could simply be due to perspective but would be greatly concerning if true. However it is difficult to know if this is just how the molecules are represented. It should be possible to plot bond lengths, torsion angles, etc. in aggregate to ensure they adopt commonly expected lengths or angle distributions (trans, gauche-, gauche+, etc.).
5. *Visualization of Results:* the images provided are only of 3D molecules, and often some part of the molecule is obscured which makes it difficult to understand what molecule was generated -- this also hinders evaluation of the previous two weaknesses. The user should include 2D versions of the molecules (perhaps in the appendix) to aid in qualitative evaluation.
6. *Description and Evaluation of Oracles:* Additionally, the paper notes that the oracles for conditioning properties were trained by the authors. A description of these models, their training, and evaluation should be included in the paper (appendix is fine).

**Overall Recommendation**.
Given the modest improvements and significant concerns about missing key features (shape based control) and uncertainties about quality of molecules, this paper cannot be recommended for acceptance. If a more thorough evaluation of the quality of the generated molecules were provided, in addition to improved visualization of the molecules, the paper could be moved to a weak accept. Development of a molecular shape oracle for controlling molecule shape and electrostatics, likely a more substantial task, would further improve the submission.

---

### Official Review · Reviewer_b1Qz · 2024-06-11
**Classifier-Free Guidance for Inverse Molecular Design**

**Rating:** 7
**Confidence:** 4

**Review:**

This paper proposes Energy-Free Guidance for Geometric Diffusion Models (EFG-GDM) and applies it to inverse molecular design on the QM9 dataset. The approach uses standard classifier-free guidance: it trains a single model to produce both conditional and unconditional score estimates via random masking of the conditioning information and composes the resulting score estimates during sampling to guide molecular generation.
Results show improved design accuracy over conditional equivariant diffusion models and energy-guided baselines on 4 out of 6 properties, with ablations confirming the benefit of joint training over separate models. The method also maintains higher molecular stability compared to energy-guided approaches.

While the adaptation of classifier-free guidance to this domain is fairly straightforward, the empirical results are promising and the simplified training and sampling procedure is appealing. In the future, a more detailed analysis of the impact of the masking probability (i.e. the ratio of unconditional vs conditional scores) and a closer look at the design accuracy vs. stability tradeoff (especially at larger guidance weights) would be informative.